# Estimating Monthly River Discharges from GRACE/GRACE-FO Terrestrial Water Storage Anomalies

**Bhavya Duvvuri *** and **Edward Beighley**

Civil and Environmental Engineering, Northeastern University, Boston, MA 02115-5005, USA;
r.beighley@northeastern.edu
* Correspondence: duvvuri.b@northeastern.edu

**Abstract:** Simulating river discharge is a complex convolution depending on precipitation, runoff generation and transformation, and network attenuation. Terrestrial water storage anomalies (*TWSA*) from NASA's Gravity Recovery and Climate Experiment (GRACE) and its follow-on mission can be used to estimate monthly river discharge (*Q*). Monthly discharges for the period April 2002–January 2022 are estimated at 2870 U.S. Geological Survey gauge locations (draining 1K to 3M km$^2$) throughout the continental U.S. (CONUS) using two-parameter exponential relationships between *TWSA* and *Q*. Roughly 70% of the study sites have a model performance exceeding the expected performance of other satellite-derived discharge products. The results show how the two model parameters vary based on hydrologic characteristics (annual precipitation and range in *TWSA*) and that model performance can be affected by snow accumulation/melt, water regulation (dams/reservoirs) or GRACE signal leakage. The generally favorable model performance and our understanding of variability in model applicability and associated parameters suggest that this concept can be expanded to other regions and ungauged locations.

**Keywords:** GRACE; *TWSA*; signal leakage; monthly discharge; exponential relation

## 1. Introduction

River discharge is an important flux, which integrates the processes and pathways connecting terrestrial water storages, such as snow, soil moisture, surface water and groundwater. Water storage, which is impacted by the time history of climate forcings and anthropogenic activities, is vital for hydropower, irrigation, domestic water supply and associated ecosystems, and it modulates streamflow behavior [1]. Given the many complexities in quantifying terrestrial water storage [2], river discharge is often used to characterize the hydrologic state of a system and as the basis for many water management decisions (e.g., reservoir releases, river withdrawals). However, in situ streamflow measurements are declining globally, especially in remote catchments and developing regions [3,4], and the relationship between water storage and discharge has been shown to vary based on catchment characteristics [5,6].

Research investigating storage–discharge relationships was first reported by Ref. [7], which described recession curves in terms of the drainage behavior for hillslope or channel storage elements. Expanded by Ref. [5], a simple dynamic system approach to create a functional relationship between discharge and effective basin water storage was proposed to model river discharge using only precipitation and evapotranspiration. Building on this concept and NASA's Gravity Recovery and Climate Experiment (GRACE), Ref. [6] presented a method for estimating discharge from GRACE terrestrial water storage anomalies (*TWSA*) [8] using an exponential relationship. The general concept of non-linear relations between catchment water storage and discharge (i.e., discharge increases/decreases with increasing/decreasing water storage) has also been used in other studies [9,10]. Additionally, researchers have leveraged the knowledge of water storage and discharge dynamics

to characterize the degree of non-linearity in the catchment response, derive streamflow statistics and improve our understanding of hydrologic behavior [11,12].

Ref. [5] focused on characterizing catchments as simple first-order non-linear dynamical systems, where the storage–discharge function can be inferred from analysis of streamflow fluctuations. The solution to the sensitivity equation (Equation (1)), which represents the sensitivity of discharge to changes in storage, is shown in Equation (3), which is derived from a water balance perspective. It has three cases, each at a different value of $b$ (i.e., the log–log slope of the best fit line for Equation (2)).

$$g(Q) = \frac{dQ}{dS} \approx \frac{dQ}{P - E - Q} \approx -\frac{dQ}{Q}|P \ll Q, E \ll Q \tag{1}$$

Of the three fluxes ($P, E, Q$), discharge can be measured more reliably than precipitation or evapotranspiration at the whole-catchment scale. The above equation implies that one can estimate the sensitivity function $g(Q)$ from the time series of $Q$ alone. Ref. [5] has noted that the above equation could be approximated in the time intervals when the precipitation and evapotranspiration fluxes are small compared to discharge ($P \ll Q$ and $E \ll Q$).

The power law relationships for streamflow recession are analytically tractable and have been widely reported since the mid-1900s, as below.

$$-\frac{dQ}{dt} = a\,Q^b \tag{2}$$

$$Q = f(S) = \sqrt[2-b]{Q_r(S - S_0)/k_1} \tag{3}$$

where $Q_r$ is an arbitrary reference discharge, and $k_1 = \frac{Q_r^{2-b}}{[(2-b)a]}$.

The solution to Equation (1) when $b$ is equal to 2 is an exponential equation (Equation (4)), which is suitable for our study, as we are using anomaly values, which are negative during dry seasons. The other two formulations (cases) where $b$ is larger and less than 2 cannot utilize negative values and are irrelevant to this study. Please refer to Ref. [5] to learn more about the other solutions.

$$Q = f(S) = Q_r\,e^{(S_e - S_0)} \tag{4}$$

Here, $S_e$ represents total water storage at time $t$, and $S_0$ represents the residual storage remaining in the catchment when discharge drops to zero; thus, $S_0$ represents the value of storage when $Q = Q_r$.

The total water storage ($S_e$) can be divided into dynamic (i.e., represented as *TWSA(t)*) and baseline or storage offset ($S_0$):

$$S_e(t) = TWSA\,(t) + S_0 \tag{5}$$

as described in Refs. [6,13]. Figure 1 shows how *TWSA* and total storage ($S_e$) can be used interchangeably, as they provide the same exponential coefficient ($\beta$) and goodness of fit ($R^2$), which accounts for the change in discharge with changing storage. However, the leading coefficient ($\alpha$) values vary with the storage offset ($S_0$) value. In this study, we estimate the $\alpha$ and $\beta$ associated with *TWSA* and river discharge. This offset cannot be measured directly but should correspond to the long-term mean water storage for the region of interest used in the postprocessing of the GRACE data to calculate the terrestrial water storage anomaly values ($S_e - S_0$). Ref. [6] attempted to estimate the drainable storage ($S_e$) from discharge storage relationships using an assumption that the baseflow is a linear function of storage.

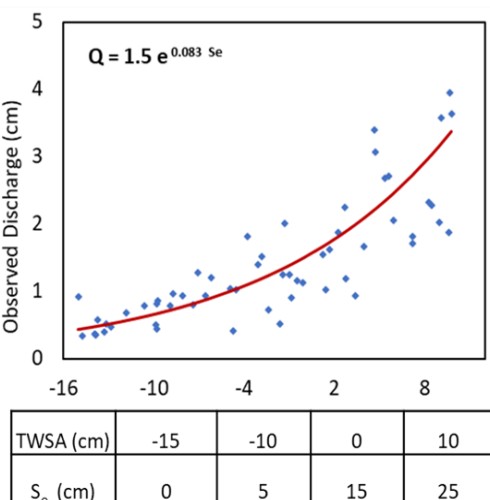

| TWSA (cm) | -15 | -10 | 0 | 10 |
|---|---|---|---|---|
| $S_e$ (cm) | 0 | 5 | 15 | 25 |

**Figure 1.** A representation of the storage–discharge relation based on Equation (5) with an assumed $S_o$ value of 15 cm; $S_e$ represents storage in GRACE *TWSA* or absolute storage (cm). The blue points represent the observed discharge in cm. The red line represents the exponential model where $\alpha$, $\beta$ is 1.5 and 0.083 respectively.

GRACE and the current follow-on mission (GRACE-FO) measure monthly changes in the earth's gravitational field, which are proportional to regional changes in total water storage [14]. The GRACE observations are ultimately used to derive monthly *TWSA*, which represents changes in the vertical sum of water from the earth's surface and subsurface storages relative to an average potential for a specified period (e.g., 2004 through 2009 for the JPL RL06_v02 product used in this study). GRACE *TWSA* have been used in many water resource applications: drought monitoring [15], characterizing drainable water in rivers and catchments [6,16], flood forecasting [17], predicting streamflow [18,19], water balance modeling [20], characterizing hydrologic signatures [21] and understanding hydrograph behavior [19,22]. Ref. [23] assessed variations in the annual runoff coefficient from a water balance equation using GRACE along with other remotely sensed and observational data. They evaluated three GRACE solutions and two TWS reconstruction products in assessing the interaction of evapotranspiration and runoff coefficients. Relevant to this study, Ref. [9] used streamflow and GRACE *TWSA* to predict the Brutsaert–Nieber equation k coefficient (i.e., the power law recession coefficient, which effectively characterizes basin-scale storage–discharge relationships) for 51 watersheds in the U.S. with drainage areas ranging from roughly 100 to 10,000 km². Similarly, Ref. [6] developed exponential relationships between *TWSA* and river discharge for 12 U.S. watersheds draining at least 200,000 km². These studies highlight the potential range in applicable basin scales for extracting hydrologic insights from GRACE *TWSA*.

Building on Ref. [6] and the potential for smaller scale applications noted in Ref. [9], this study develops non-linear exponential models for storage–discharge relations [$Q = \alpha e^{TWSA}$] for 2870 gauged watersheds distributed throughout the continental U.S. (CONUS), where storage is represented by GRACE/GRACE-FO *TWSA*, and discharge ($Q$) is measured at U.S. Geological Survey (USGS) gauging stations. The intent of this study is to assess the applicability of a remote-sensing-based approach for estimating river discharges. Here, we focus on an application of the storage–discharge relationship at gauged locations. Future research will leverage the findings from this study to develop the methods for estimating the required model parameters at ungauged locations.

Given that snow accumulation/melt and water regulation can impact the storage–discharge relationship (i.e., discharge does not respond to changes in storage) [6], the relationships are developed for each watershed using a combination of months and lag between *TWSA* and streamflow data pairs. The exclusion of particular months of data and/or lagging of *Q–TWSA* data pairs are intended to limit the impacts of snow accumu-

lation/melt and water regulation. To determine the optimal relationship at each site, an idealized or baseline model (12 months of data per year with no lag between *Q* and *TWSA*) is compared to the various combinations of data months and lags (i.e., test models) using the Kling–Gupta efficiency (*KGE*). A sensitivity analysis is performed to assess variability in model parameters based on the selected months of data and lag. The results show the spatial distribution of optimal model performance (*KGE*) and the associated model parameters ($\alpha$ and $\beta$).

## 2. Materials and Methods

### 2.1. Study Area

The study area covers the CONUS. The streamflow and reservoir storage time series were obtained from the USGS's National Water Information System [24,25]. Figure 2 shows the USGS gauge locations, where the *Q*–*TWSA* relationships are analyzed, and the hydrobasins are obtained from the MERIT Hydro database. Note that some of the hydrobasins extend to Mexico, Canada, Alaska, Hawaii and other islands but are not shown here. There are 8 hydrobasins in the CONUS, numbered 71–78; here, Hydrobasin 76 has no USGS gauges and is excluded. Hydrobasin 74 is the largest, as it includes the Mississippi watershed draining into the Gulf of Mexico. This hydrobasin has a mostly humid subtropical climate with varied levels of precipitation and isolated semi-arid regions. The Mississippi River basin represents the larger drainage area in the study region (3M km$^2$). Within the Mississippi, its largest tributary, the Missouri River, is impacted by snow processes and highly regulated. Hydrobasin 73 covers the east coast, which has a somewhat north–south gradient in temperature and precipitation [26] with snow effects in the Appalachian Mountains and New England regions, which receive 100–250 cm of snow annually. Hydrobasin 75 includes New Mexico and Texas and generally has a semi-arid climate with some subtropical regions, especially in Texas. This region includes the Rio Grande, which drains into the Gulf of Mexico. Snowfall throughout this region is less than 10 cm per year, but in select catchments, it can be up to 50 cm. As is typical for semi-arid regions, many of its rivers are highly regulated. The watersheds in Hydrobasin 77 generally have arid to semi-arid climates toward the east side of the region and a Mediterranean climate in the California portion. The watersheds include major rivers, such as the Colorado and Sacramento Rivers draining into the Pacific Ocean. The mountainous areas within this region tend to be snow dominated. Hydrobasin 78 consists of the Columbia River draining into the Pacific Ocean. Hydrobasin 72 includes the Great Lakes' basins draining to lakes Huron, Michigan, Eerie, Superior and Ontario. In the Results and Discussion sections, select USGS gauges are used to highlight the findings related to climate (e.g., snow impacts), water regulation and proximity to the Great Lakes, which are common to many gauges within their respective hydrobasins.

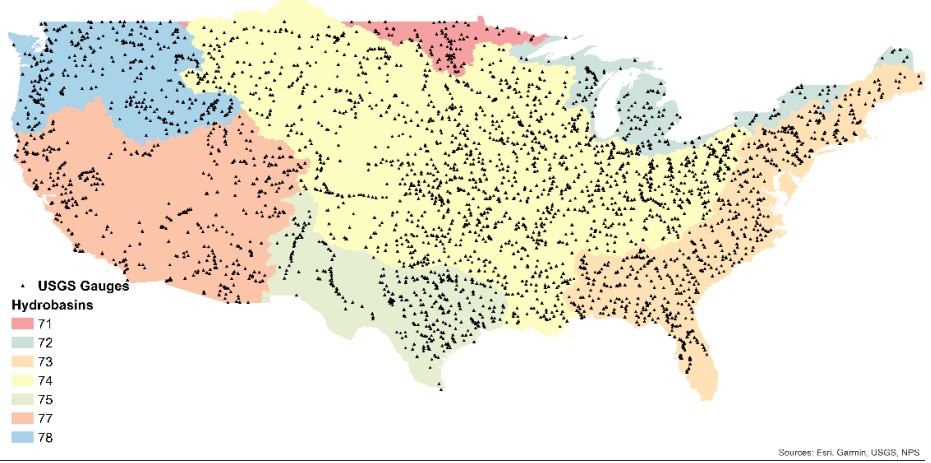

**Figure 2.** Study region showing USGS gauge locations and MERIT hydrobasins.

## 2.2. Data

The data used in this study include river networks, associated catchment boundaries and hydrologic characteristics (i.e., river discharge, lake level/storage, TWSA, snow cover, precipitation, temperature). The catchment boundaries and river reaches were obtained from the MERIT Hydro database [27,28]. The key remotely sensed data products include JPL's monthly GRACE TWSA in equivalent water thickness units gridded at 0.5 degrees (JPL RL06_v02) [29]. The version of the data used employs a coastal resolution improvement (CRI) filter, which reduces leakage errors across the coastlines. The data are available from April 2002, with missing values in 2017–2018 (11 months) due to the gap between the GRACE and GRACE-FO missions, as well as periodic missing months. In this study, the missing months are excluded, as only data pairs (*TWSA* and river discharges) are required. Note that while the resolution of the TWSA data product is $0.5 \times 0.5$ degrees, the native resolution is roughly $3 \times 3$ degrees. Here, we use the gain factors, as suggested for hydrology studies, to extrapolate to the finer scale of the data product [30,31]. For monthly snow cover, daily MODIS snow cover version 6 (MOD10CM) from the Terra satellite was used [32]. Monthly averages were determined from daily snow cover observations after filtering out days in which the percent of clear sky (no clouds) was greater than 100 to remove low-magnitude snow cover fractions to avoid the effects of erroneous snow detections. The product has a 0.05-degree spatial resolution. For monthly precipitation, the Integrated Multi-satellitE Retrievals for GPM (IMERG) precipitation version 6.1 with a 0.1-degree spatial resolution was used [33]. An approach for infusing monthly gauge information into the satellite-derived precipitation estimates is used to calculate IMERG precipitation data. All the full-resolution half-hour multi-satellite estimates over a month are summed to create a monthly multi-satellite product. Here, we used the Final Run dataset (3B-MO). Daytime land surface temperature (LST) data were obtained from the Terra MODIS Version 6.1 Level-1B (L1B) products. The LST product (MOD11C3) provides monthly composited and averaged temperature and emissivity [34] values at 0.05-degree spatial resolution in Kelvin. In addition to LST, potential evapotranspiration (PET) was calculated using the MODIS LST based on the Thornthwaite method, as described in Ref. [35]. PET estimates were determined for each month, considering the month as being 30 days long, with 12 theoretical sunshine hours per day. Here, PET is intended to capture the spatial variability in potential water stress derived from remotely sensed datasets (e.g., P/PET). Future efforts will explore more physically based ET estimates. The reservoir storage time series were obtained from USGS's National Water Information System [24,25]. The lake water levels for the Michigan–Huron lake were obtained from the NOAA website [36].

It is important to note that there are multiple GRACE TWSA solutions and sources of precipitation and temperature data available. In this study, only one GRACE data product and one source of precipitation and temperature data are used. Building on Refs. [37,38], future studies will explore using multiple sources of GRACE TWSA and supporting data products to enable robust uncertainty assessment and assessment of when and where the different GRACE solutions perform best.

## 2.3. Data Processing

River discharge at a location and the corresponding water storage within the drainage areas (i.e., catchments) upstream of the location are central to this study. For every river reach, area weighting was used to determine the average hydrologic quantities (precipitation, snow cover, temperature, PET, *TWSA*) for all upstream catchments draining to each reach. For all hydrologic quantities listed in the Data subsection, spatially averaged values were determined for all river reaches in the CONUS draining at least 1000 km². Using the gauge locations reported by the USGS, gauges were linked to river reaches. Gauges with reported drainage areas not within $\pm 1.25$ times the river-network-derived drainage area or with fewer than 100 *TWSA*–discharge data pairs (for streamflow data starting from 2002) were eliminated. The discharge time series were converted to cm/month units using the

drainage area of the gauge. The resulting database contains monthly time series values of spatially averaged hydrologic quantities (e.g., P, T, PET, *TWSA*, etc.) and monthly river discharges for the period April 2002–January 2022. GRACE *TWSA* has two values reported for January 2012 and April 2015. The *TWSA* values reported for 1 January 2012 are used as the missing December 2011 values. The values reported for 31 April 2014 are excluded.

### 2.4. Model Development and Selection

An exponential model was fitted to monthly discharge and *TWSA* data pairs using

$$Q = \alpha \, e^{\beta \, TWSA} \tag{6}$$

where $\alpha$ and $\beta$ are the model parameters optimized for each gauged watershed. In total, 219 models were created for each study site based on unique relationships developed using a combination of 1–12 months of *Q*–*TWSA* data pairs within a given year (e.g., the periods March–November or April–October include 9 months of data from each year) and 0 or 1 month of lag between *TWSA* and streamflow (i.e., *TWSA* in January paired with discharge in February would be 1 month of lag) series. In this study, the data pairs used to develop the model are referenced as "training data".

The optimal model parameters were determined by maximizing the Kling–Gupta efficiency (*KGE*) [39]. *KGE* accounts for three components: mean bias ratio, variability and correlation between the observed and simulated discharge series. *KGE* is defined as

$$KGE = \sqrt{\left(1 - \frac{\mu_{sim}}{\mu_{obs}}\right)^2 + \left(1 - \frac{\sigma_{sim}}{\sigma_{obs}}\right)^2 + (1 - R)^2} \tag{7}$$

where $\frac{\sigma_{sim}}{\sigma_{obs}}$ is the flow variability; R is the correlation coefficient; and $\frac{\mu_{sim}}{\mu_{obs}}$ is the bias ratio between simulated and observed discharges. For the mean flow benchmark—where all of the model's simulated values are equal to the mean (bias ratio = 1), which makes flow variability ($\sigma_{sim}$) and correlation zero—the *KGE* is equal to $-0.41$. Thus, a *KGE* greater than $-0.41$ indicates that the selected model performs better than the mean discharge prediction. A perfect model has a *KGE* of 1 when the simulated discharge is identical to the value of observed discharge (three components are unity). There is no established threshold for good or bad *KGE* (or mean baseline performance), and it depends on the modeler's discretion. If a threshold is set midway between the best performance (*KGE* = 1) and mean flow benchmark (*KGE* = $-0.41$), the benchmark for good models is 0.30. We set our threshold for good models at 0.32, based on the arguments in Ref. [40]. The author considers a threshold of 0.32 for a "good" *KGE*, which is estimated by a correlation coefficient of 0.5, a bias ratio of 0.83 and a relative variability of 0.58, based on the expected performance of discharge algorithms for NASA's forthcoming Surface Water and Ocean Topography (SWOT) mission [40–43].

In this study, three *KGE* values are referenced (baseline, training and test). The *KGE* for the models trained with all months of data and no lag is referred to as the baseline *KGE*. The *KGE* for optimized models developed from the training data is called the training *KGE*. The *KGE* for optimized models from the training data evaluated using all the data available and no lag is called the test *KGE*. The optimized or "best" model at each site was selected based on the model with the largest test *KGE*, which was greater than its corresponding baseline *KGE* and less than its corresponding training *KGE*.

### 2.5. Model Applicability

Remotely sensed datasets representing precipitation, land surface temperature, snow cover and *TWSA* variables play an important role in determining the applicability of the proposed non-linear model. The correlation of *TWSA* with precipitation and snow cover, trends in precipitation and *TWSA* over time, and magnitude of precipitation relative to potential evapotranspiration (P/PET) (determined using the Thornthwaite [35] formula

and remotely sensed land surface temperature) have high feature importance among all the variables analyzed. These features are used to develop a decision tree to determine whether the streamflow at a given location can or cannot be approximated using a *Q–TWSA* relation. For this analysis, a *KGE* ≥ 0.32 is assumed to provide an applicable *Q–TWSA* relation.

*2.6. Regionalization*

To explore the spatial patterns for simulated discharges, the observed river discharges and optimized model parameters (*α*, *β*) from the gauge locations are propagated throughout the study region's river networks. From the measured discharge series, we only propagated the mean monthly discharges. Each river reach was assigned upstream and downstream gauges within the river network. The discharges from the immediate upstream or down-stream gauges were used to estimate the discharge based on drainage area ratios and area weighting [40]. If a river reach had multiple gauges upstream and a downstream gauge, an average of area-scaled discharges from all the gauges upstream was determined and used to estimate an area-weighted average with the downstream gauge value.

To transfer the model parameters (*α*, *β*), which were then used to produce monthly discharges (cm/month), area weighting was not used. Rather, the parameter values deter-mined for each gauge location were simply moved upstream and downstream within the network. If a river reach had a gauge upstream and downstream, an area-weighted average was used, producing parameter values constrained between the two gauge-derived values and approaching the gauge values along the reaches closest to a given gauge. For future studies, we will explore additional methods of propagating model parameters with more sophisticated regionalization techniques. Using the transferred model parameters, monthly discharges were determined based on the GRACE *TWSA* time series. For comparison with observed discharge, mean monthly discharges were determined from the simulated time series. However, it should be noted that the resulting differences in mean monthly discharge in each reach include both model uncertainty and the propagation approach for both observed discharges and the developed model parameters, which does not account for human activities (e.g., dams/reservoirs, withdrawals) along a river. Future research will explore the propagation methods, which take these activities into account.

## 3. Results

The relationships between *TWSA* and monthly river discharge for 2870 watersheds draining to USGS streamflow gauges throughout the CONUS were generated for the period 2002–2020. Figure 3 shows the results for three gauges, which highlight the varying levels of model performance. Given the potential impacts of reservoir operations and snow accumulation/melt, relationships using various combinations of monthly *Q–TWSA* data pairs (e.g., May–October only) and 0–1 month of lag between *Q–TWSA* pairs were explored. The optimal relationship from the 219 potential combinations were selected based on training, testing and baseline *KGE* (see the Materials and Methods section for additional information). For reference, the baseline model is developed using no lag and all 12 months of data pairs from each year.

Figure 3 shows the resulting exponential *TWSA–Q* relationships and corresponding measured and modeled discharge series for three gauges: USGS ID 9144250, Gunnison River at delta, CO, Hydrobasin 77 (Figure 3a,d); USGS ID 1633000, Shenandoah River at Mount Jackson, VA, Hydrobasin 73 (Figure 3b,e); and USGS ID 13266000, Weiser River near Weiser, ID, Hydrobasin 78 (Figure 3c,f). The selected gauges highlight the typical model performance for various *KGE* testing values. Note that a *KGE* greater than −0.41 indicates that the selected model performs better than the mean discharge [24], and a value of roughly 0.32 is typical for remotely sensed discharge estimates [41]. A perfect model has a *KGE* of 1. More discussion on *KGE* is provided in the Materials and Methods section. Overall, the model performance at 67% of the sites results in a *KGE* greater than 0.32, with 100% having a *KGE* greater than −0.41. Figure 4a shows the spatial distribution of model performance (*KGE*). In general, good model performance is achieved (*KGE* > 0.32)

throughout the CONUS, with a few obvious clusters of lower performance near the Great Lakes, the Midwest and central Florida.

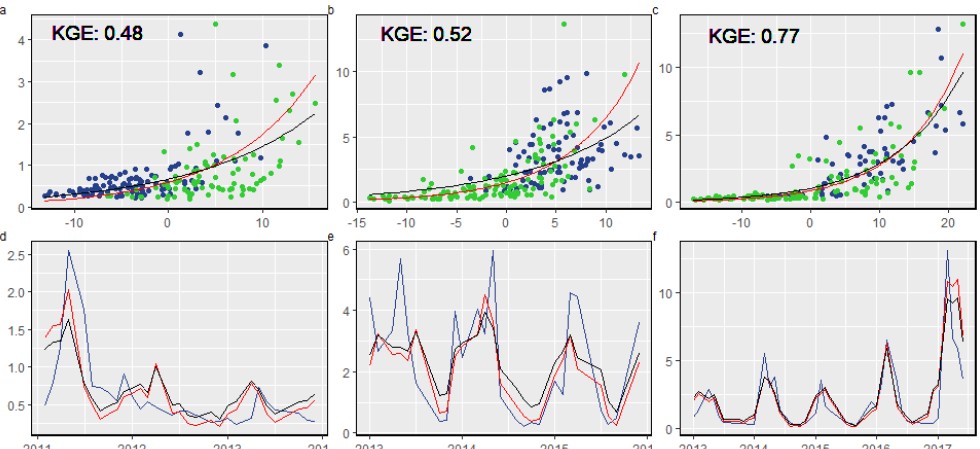

**Figure 3.** (**a**–**c**) Relationships between streamflow (cm/month, *y*-axis) and *TWSA* (cm, *x*-axis) and (**d**–**f**) comparisons between predicted and observed streamflow; columns from left to right show the results for sites with *KGE* testing values of 0.48, 0.52 and 0.77, respectively. (**a**–**c**) The red and black lines represent the optimized and baseline models, respectively. The points represent the available *TWSA–Q* data pairs, with the green points corresponding to the data pairs used to train the optimal model. (**d**–**f**) The blue, red and black lines represent the observed and predicted data of the optimized model and baseline model (trained in all 12 months), respectively.

In terms of watershed scale, the model performance only varies slightly with drainage area (i.e., 66% vs. 76% of sites draining ≤3200 km² vs. ≥320,000 km² have a *KGE* greater than 0.32). Specific to the GRACE *TWSA* data resolution, 65% of gauges draining ≤2500 km² (0.5 × 0.5 degrees; data product resolution) have a *KGE* greater than 0.32, and 66% of gauges draining ≤90,000 km² (3 × 3 degrees; native resolution) have a *KGE* greater than 0.32. For reference to the larger GRACE scales, 77% of gauges draining ≥90,000 km² have a *KGE* greater than 0.32. While a smaller fraction of gauges have favorable model performance at scales lower than the native GRACE resolution, the overall percentage of gauges with favorable model performance is encouraging, suggesting that watershed scaling of sub-grid *TWSA* can be performed. For additional information on the justification of applying GRACE (coarse resolution) on a smaller scale, see Supplementary Information Figure S1.

In snow-covered regions and highly regulated regions, the difference between the training and testing *KGE* is large. Maps of the training, testing and absolute *KGE* difference are provided in Supplementary Information Figure S2. Overall, 97% of the gauges have an absolute *KGE* difference of less than 100%. The number of gauges with a *KGE* difference greater than 30% increases with an increase in the mean snow cover (%); a 20% increase in the number of gauges was observed at between 30 and 60% of snow cover. This pattern was also observed in gauges with a *KGE* difference between 10 and 30%. Caution should be taken when applying these models to excluded months in such regions. There is a negative slope for *KGE* vs. snow cover (%), and the negative slope becomes steeper when only the gauges affected by the snow (greater than 30%) are considered. In general, the number of gauges with a *KGE* of less than 0.32 increases with snow cover, especially between 30 and 60% of snow cover (%). The *KGE* of the excluded months decreases with an increase in the snow cover, especially in the range mentioned above.

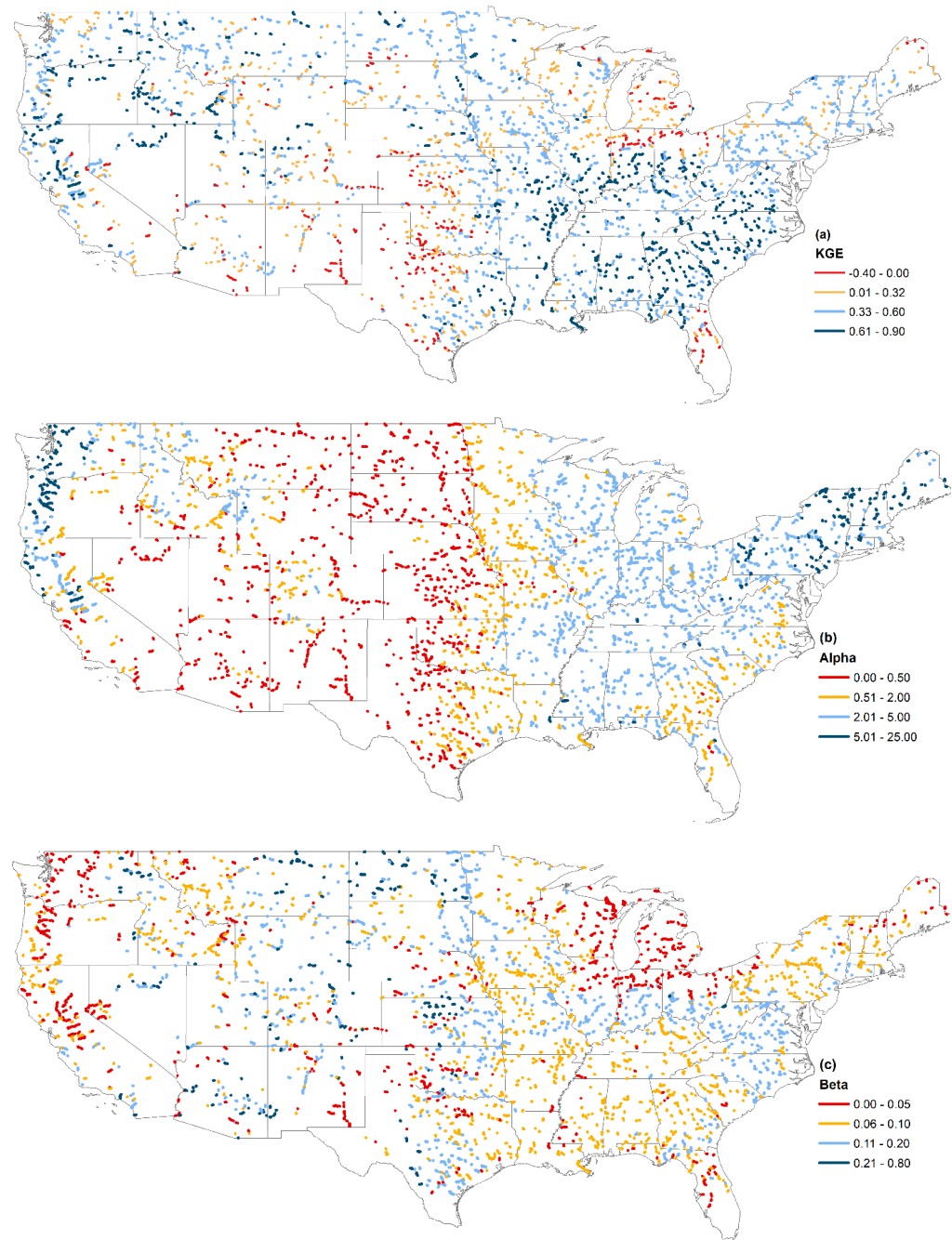

**Figure 4.** Spatial distribution of (**a**) *KGE*, (**b**) alpha ($\alpha$) and (**c**) beta ($\beta$) throughout the CONUS. The results are shown for 2533 sites, which have at least 8 years of data and beta greater than 0. The *KGE* shown represents the testing *KGE* based on optimized alpha and beta model parameters.

### 3.1. Optimized Models

The leading model coefficient ($\alpha$) has a non-uniform spatial distribution ranging from near 0 to 25, with 76% of the sites having a value less than 3 (Figure 4b). There are some discernible clusters, which tend to vary based on streamflow. For example, a comparison of $\alpha$ and annual streamflow shows a strong linear relationship ($R^2 > 0.9$) for gauges with *KGE* > 0.32. Given that streamflow and precipitation have similar spatial patterns in most regions, except in the more arid regions (e.g., southwest regions) and regions with higher seasonal snow cover, we looked at the relationships between $\alpha$ and precipitation for the potential to estimate $\alpha$ at ungauged locations. The correlation between $\alpha$ and median annual precipitation is 0.6 for the sites with testing *KGE* > 0.32. Given the high degree of

correlation with precipitation, $\alpha$ was found to vary exponentially with precipitation (mean and median annual P), with a coefficient of discrimination ($R^2$) of 0.41 (Figure 5a).

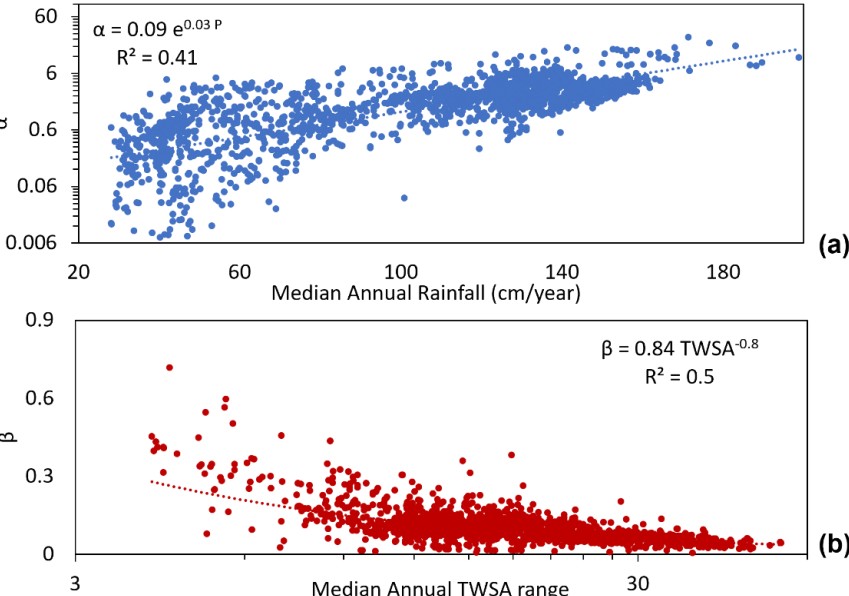

**Figure 5.** Relationships between model parameters and hydrologic variables: (**a**) scatter plot of alpha, $\alpha$, vs. median annual precipitation (cm/yr at gauges in the study. The line represents the alpha-median annual rainfall relation displayed. and (**b**) scatter plot of beta, $\beta$, vs. median annual range in *TWSA* (cm) at gauges in the study. The line represents the beta-median annual *TWSA range* relation displayed.

The exponential coefficient ($\beta$) also has a non-uniform spatial distribution and is found to have high values at some sites in Arizona, the Pacific Northwest and Arkansans River watersheds. The beta varies between 0 and 0.8, with 90% of the sites having a value between >0 and 0.17. The beta values vary by watershed/region but remain relatively consistent along a given river. For example, along a 225 km stretch of the James River flowing into the Missouri River, the $\beta$ values are relatively constant, ranging from 0.13 to 0.14 for five sites with draining areas increasing from 36,800 to 55,000 km$^2$. This is consistent with findings that $\beta$ varies with the range in *TWSA* as an exponential decay relation. For example, $\beta$ is associated with the median annual dispersion in *TWSA* ($R^2$ = 0.5) at sites with testing *KGE* > 0.32 (Figure 5b). Given the spatial scale of GRACE *TWSA* and the drainage network processing used here, the *TWSA* signals tend to be similar along larger rivers, leading to similar $\beta$ values and increasing $\alpha$ values to account for the increase in discharge, as the drainage area increases along the river.

*3.2. Sensitivity Analysis*

The optimized or "best" model was selected using the procedure described in the Materials and Methods section. Here, we investigate the differences in model parameters and performance between the optimized, baseline and an ensemble of top-performing models. Figure 6 shows the ratios of the absolute difference between optimized and baseline model values (e.g., *KGE*, $\alpha$, $\beta$) divided by the optimized model values as gray bars. Lower ratios indicate that the optimized and baseline values are similar. The $\alpha$ ratio indicates that the optimized $\alpha$ values are within 30% of their baseline $\alpha$ at 87% of the sites. Similarly, the $\beta$ ratio indicates that the optimized $\beta$ values are within 30% of their baseline $\beta$ at 69% of the sites. Given that $\alpha$ and $\beta$ can vary in different directions (e.g., smaller $\alpha$ and larger $\beta$ relative to baseline values), the resulting variability in predicted mean discharges is muted. For example, the predicted mean discharges from the optimized and baseline models vary by less than 30% at 99% of the sites. A consistent finding is shown for *KGE*, with

76% of the sites having a *KGE* ratio of less than 30%. The similarity between the baseline and optimized model performance is especially encouraging for future applications in ungauged locations.

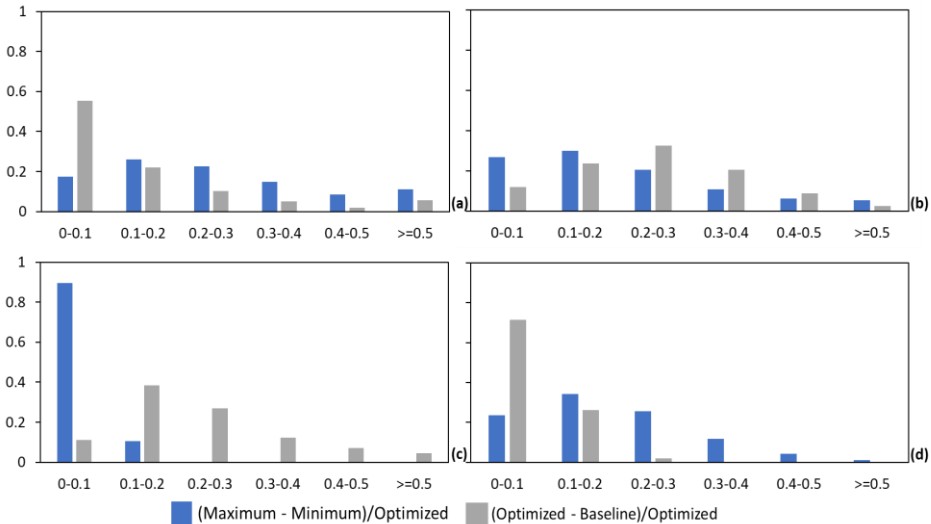

**Figure 6.** Variability in model parameters: (**a**) $\alpha$ and (**b**) $\beta$ model performance, (**c**) *KGE* and (**d**) mean discharge for the optimized (i.e., best), baseline and selected top-performing models; variability shown as a ratio of absolute difference/range in model values divided by the optimized model value; *KGE* for optimized model representing testing *KGE*.

In the above figure, we investigated the optimized models relative to the baseline models. Here, we look at variability in a subset of models with similar performance. In general, the optimized models selected were not sensitive to the model selection procedure. Models with a test *KGE* within at least 90% of the optimized models' test *KGE* are selected as "top models" at each site. Figure 6 shows the ratios of dispersion (maximum-minimum) in the values from the top model ensemble (e.g., *KGE*, $\alpha$, $\beta$) divided by the optimized model value as blue bars. Small ratios correspond to a small difference between the maximum and minimum model values in the subset relative to the optimized model value. In Figure 6a, roughly 66% of the sites have an $\alpha$ ratio of less than 30%. The $\beta$ ratio shows slightly less variability, with 77% of the sites having a ratio of less than 30% (Figure 6c). As in the above discussion, the variability in the mean of the predicted discharges from the top models is muted; 83% of the sites have ratios of less than 30% (Figure 6d). As in the comparison with the baseline model, the limited variability within the top-performing models suggests that the model selection approach does not overly influence the optimized model values or overall model performance.

The above results suggest that the model parameters and performance within the top-performing ensemble of models do not vary significantly. Figure 7 compares the relationships between discharge and *TWSA* for one selected model ensemble (USGS Gauge ID 4260500). The model ensemble essentially captures the range between the baseline and optimized models. The key difference between the baseline and the optimized model is the rate of increase in discharge with increasing *TWSA* (i.e., larger $\beta$ value). The optimization process tends to reduce the smaller discharges (Figure 3), which increases the relative weight of the larger discharges in the fitting process. Because all models use *TWSA* to estimate the discharge, the differences between the models are shown as vertical shifts in discharges (Figure 7b), with the optimized model estimating larger high flows and smaller low flows.

In terms of performance over a range of discharges, the optimized models tend to perform better with low and high flows as compared to the baseline model (Figure 8a,b), but the baseline model performs better for median and mean discharges (Figure 8c,d).

While the baseline model tends to have slightly more variability (i.e., the baseline points are more scattered than the optimized model points in Figure 8), the differences are more noticeable in terms of bias. The bias ratios in the minimum, maximum and median values are nearly 1, ranging from 0.6 to 1.5 across all models.

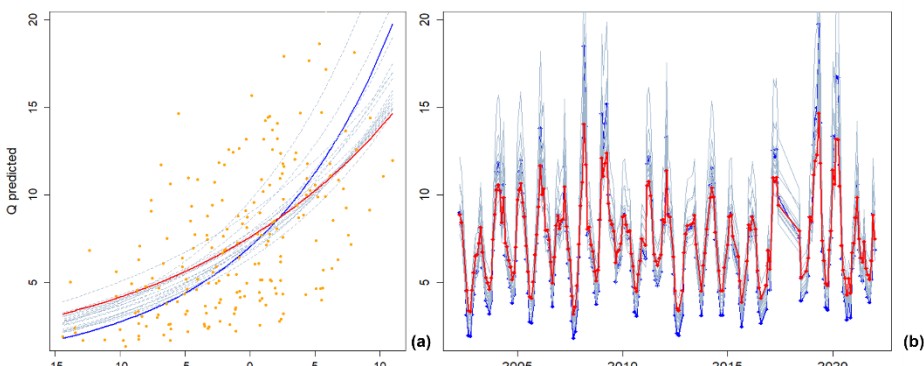

**Figure 7.** (**a**) Relationship between *TWSA*–discharge and (**b**) simulated discharge series; optimized model (dark blue), baseline (red) and top models (gray dashed lines); *x*-axis values for *TWSA* are in cm.

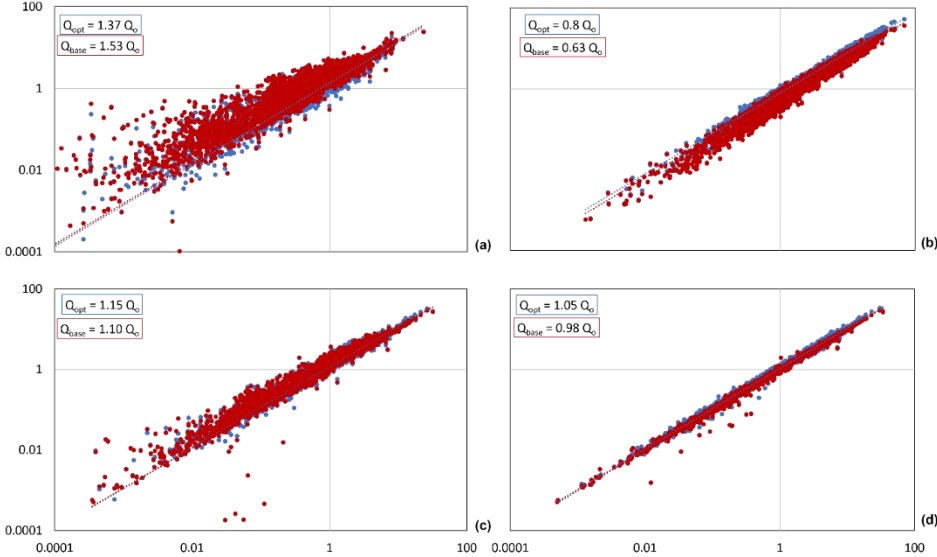

**Figure 8.** Mean annual (**a**) minimum, (**b**) maximum, (**c**) median and (**d**) mean predicted discharge (cm/month, *y*-axis) compared to in situ discharges (cm/month, *x*-axis). The red points correspond to the baseline model simulations; the blue points correspond to the optimized model simulations.

### 3.3. Model Applicability

A simple classification tree with average annual precipitation relative to potential evapotranspiration (P/PET), correlation of *TWSA* with precipitation and snow cover is adequate for predicting model applicability (Figure 9). While the results shown in Figure 9 are preliminary, the key performance metrics suggest that the proposed modeling approach can be applied to ungauged locations using remotely sensed data to define the feature values. Here, we use four error metrics (accuracy, false alarm rate, probability of detection and bias) to evaluate the ability of our preliminary decision tree (Figure 9) to identify locations for which the proposed non-linear *Q*–*TWSA* model approach is likely to yield good performance (*KGE* > 0.32). A "positive" classification or outcome indicates that the decision tree logic suggests that the observed *Q*–*TWSA* relationship resulted in good model performance (*KGE* > 0.32); a "negative" classification is poor performance (*KGE* < 0.32). The accuracy, defined as the number of correct classifications (true positives = 1578 and

true negatives = 419) divided by the total number of classifications (2502), is 0.8, with 1.0 being ideal. The false alarm ratio, defined as the number of false positives divided by the total number of logic tree positive classifications (1956), is 0.19, with 0 being ideal. The probability of detection or hit score, defined as the number of true positives divided by the total number of observed *Q–TWSA* relationships with positive outcomes (1728), is 0.93, with 1.0 being ideal. The overall bias, defined as the total number of logic tree positive classifications divided by the total number of observed *Q–TWSA* relationships with positive outcomes, is 1.15, with 1.0 being ideal. Collectively, the high accuracy (80%) and hit score (93%), combined with a reasonable false alarm ratio (19%) and bias 1.15, suggest that the preliminary decision tree can be used to identify locations where the proposed non-linear *Q–TWSA* modeling approach is likely to perform well. This is a key finding because the decision tree uses relevant hydrologic quantities derived from remotely sensed datasets, which can be generated globally.

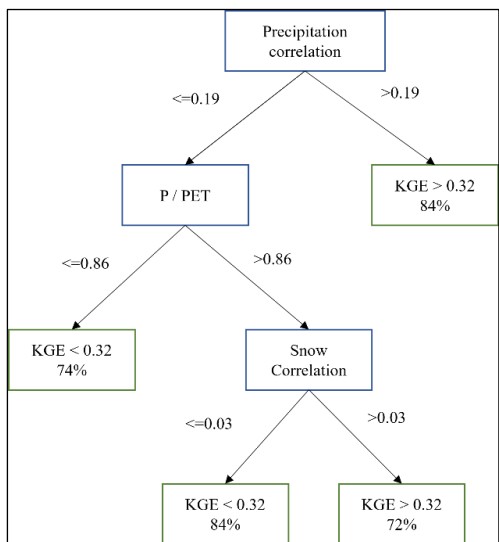

**Figure 9.** Decision tree for predicting locations with good (*KGE* > 0.32) model performance; leaf nodes are green, and decision variables are shown in blue boxes.

Overall, the modeling approach works well throughout the CONUS, with the exception of watersheds, where the rainfall–runoff processes are highly non-linear (semi-arid); the streamflow is regulated (Midwest); there is a strong snow accumulation/melt signal (upper Midwest); groundwater depletion and/or changes in lake storage have a long-term trend (lower Midwest); or the GRACE signal is likely impacted by leakage effects (Great Lakes). In terms of potential applications in ungauged locations, the challenges mentioned above can likely be identified using remotely sensed datasets.

### 3.4. Regionalization

To expand the assessment from individual gauge locations to the CONUS-wide river network, we regionalized the gauge location values (observed mean annual discharge and optimized model parameters: $\alpha$, $\beta$; see the Materials and Methods section). Figure 10 shows the comparisons. The regionalized observed mean monthly discharges are shown for all the river reaches draining an area greater than 1000 km$^2$ (Q1) in Figure 10a. The simulated mean monthly river discharges (i.e., Q2, based on regionalized model parameters: $\alpha$, $\beta$) are shown in Figure 10b. In general, the network-wide river discharges are similar. The differences are shown in Figure 10c, where the predictions (cm/month) are over- or under-estimated mostly in regions (eastern and Pacific Northwest) where the monthly discharges are large (e.g., >5 cm/month). The predictions in the semi-arid, arid and Midwest watersheds vary from the observed discharges by only ±0.5 cm/month. At 74% of the catchments, the percentage difference is within 25%. The discharges are over-

estimated in 20% of catchments and under-estimated in 7% of catchments (±25% percent change being considered baseline). Overall, the mean monthly discharges along roughly 1M km of river reaches were predicted within ±25% of the in situ observations. While the regionalization shown here only considered river reaches, which had gauge locations upstream and/or downstream, the favorable comparisons suggest that the additional regionalization techniques (e.g., hydrologic similarity) may be possible for river reaches, which do not contain gauges upstream and/or downstream. It should also be noted that the regionalization of model parameters does not take human activities along the river network into account. Thus, the resulting differences include both model uncertainty and the potential impacts of water regulation.

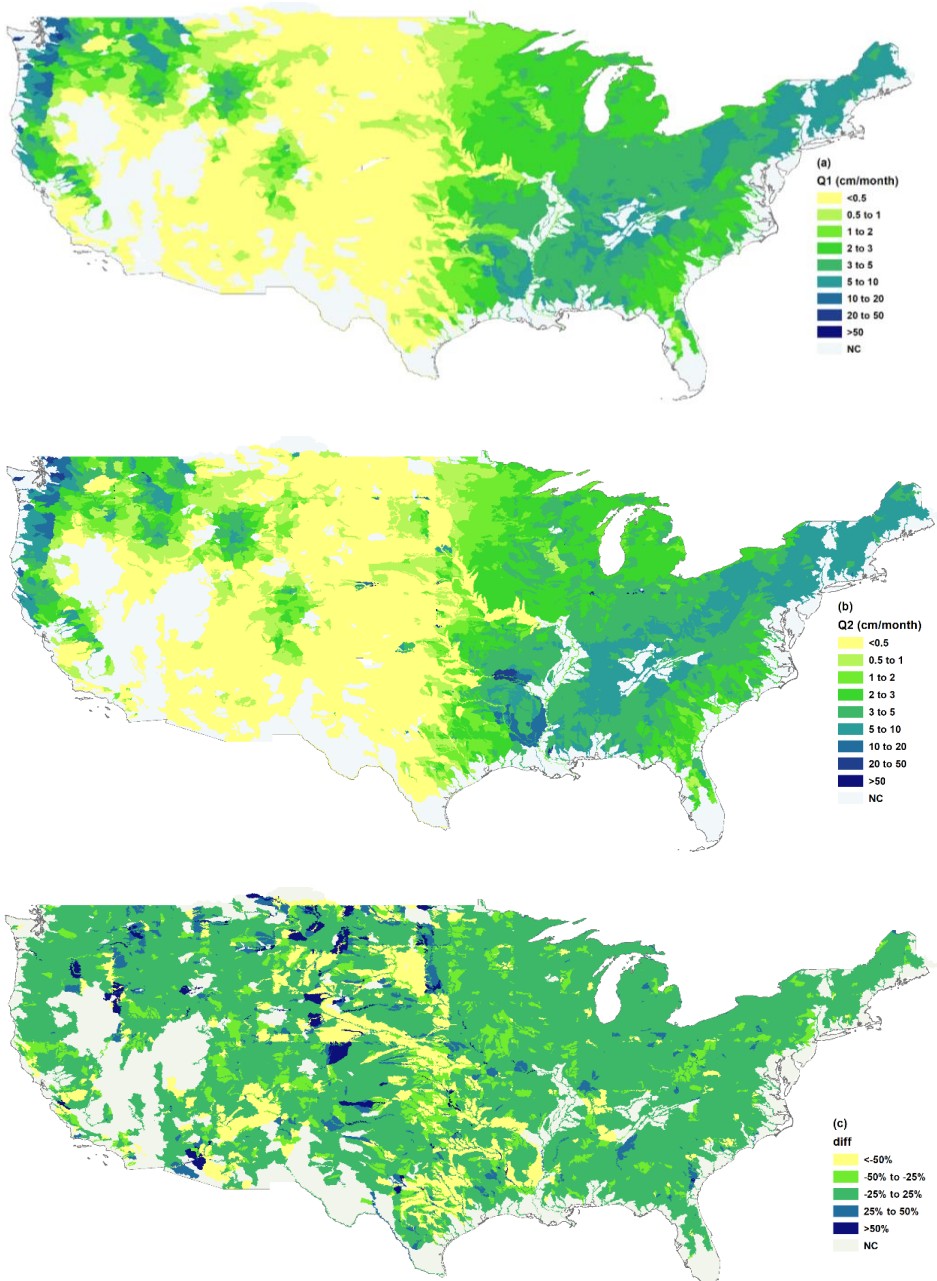

**Figure 10.** Mean monthly river discharges for river reaches draining an area greater than 1000 km$^2$ showing (**a**) observed (Q1) and (**b**) predicted (Q2) discharges based on regionalization and (**c**) differences between observed and predicted discharges. NC corresponds to regions, where streamflow was not observed or predicted.

## 4. Discussion

Based on the established storage (or GRACE *TWSA*)–discharge relationships, functional exponential models [$Q = \alpha e^{\beta \, TWSA}$] were developed at USGS gauge locations throughout the CONUS. The models were optimized for the number of months and lag in the *Q–TWSA* data pairs. The intent of excluding select months and including potential lag was to (i) limit the potential impacts of natural (i.e., snow accumulation/melt) and engineered (i.e., reservoir regulation) controls, which did not follow the proposed conceptual model, and to (ii) identify rational model parameters ($\alpha$, $\beta$), which could potentially be generalized based on hydrologic characteristics for use at ungauged locations. The filtering of data tends to reduce the number of lower discharges (Figure 3a–c), adding more weight to larger values in the fitting process, which tends to increase larger discharges and decrease lower discharges as compared to the baseline model (Figure 3d–f).

The differences in model performance for the baseline, optimized and an ensemble of top-performing models are found to be similar (Figure 6). The key findings from the sensitivity analysis are as follows: (i) the optimization approach does not drastically change the model parameters or performance; and (ii) the models tend to produce reasonable discharges but with a bias toward over-estimating low flows (by 37%) and under-estimating high flows (by 20%). These are important findings, which suggest that the approach may be applicable in ungauged locations, where optimization methods will not be possible (i.e., assuming baseline model conditions; all 12 months with no lag), and methods are being developed to generalize the model parameters ($\alpha$, $\beta$).

Catchment hydrology studies investigating storage have reported that highly connected catchments [12,44,45] have straightforward relations between the discharge and storage. The storage–discharge relation is linear or quasi-linear in quick recession periods and non-linear in slow recession periods [46]. We calibrated models at the monthly scale to avoid the effect of these diurnal discharge fluctuations [47]. The hillslope riparian connectivity increases with the drainage area [44], suggesting that the complexity/connectivity increases with the drainage area, where the non-linear relationship is only present between *Q* and *TWSA* for some percentage of the water year. This contrasts with simple catchments, where the exponential relation exists throughout the year, which is also observed in Ref. [12]. This supports our approach of optimizing the number of months excluded from the fitting process.

The model performance depends on variability in the hydrologic cycle and the potential impacts of snow processes and water regulation. A key indicator of good model performance is annual precipitation and the associated streamflow. Regions with higher annual precipitation (e.g., east coast, Pacific Northwest) tend to have the best performance as compared to dryer regions (e.g., Midwest, Great Basin). For example, if the median annual precipitation is less than 50 cm/yr, the *KGE* is less than 0.32 for 30% of the sites, but if the precipitation exceeds at least 110 cm/yr, the *KGE* is greater than 0.32 for 82% of the sites. Poor model performance in dryer, semi-arid regions is likely due to the flashiness (i.e., highly non-linear rainfall–runoff behavior) of those systems or the higher degree of water regulation to maintain water supplies. In wetter regions, we also observe larger variability in *TWSA* (i.e., the difference between the minimum and maximum values each year), which enables the models to simulate larger ranges in discharge. Additionally, when a discharge series only has a few large event peaks, the models tend to predict high flows well but consistently over-estimate the lower flows. For example, if a site's discharge series coefficient of variation (CV) is greater than 6 (i.e., the standard deviation is 6 times the mean), the optimal model's *KGE* value is always less than 0.32.

Understanding the hydrologic processes captured by each model parameter can help develop the relationships for estimating the parameter pairs. The $\alpha$ value drives the mean discharge signal, with variations in *TWSA* driving the discharge up or down based on the magnitude of the corresponding $\beta$ value. While the preliminary relationships for $\alpha$ and $\beta$ shown in Figure 5 are reasonable, additional research is needed to generalize the parameter values for applications in ungauged locations. The resulting errors in the regionalized

discharges based on transferring the model parameters through the network are within ±25% of the transferred observed discharges for 74% of all river reaches used in this study (roughly 997,000 km of river reaches). These results are promising for the potential expansion to other regions and ungauged locations.

### 4.1. Model Parameters' Rationale

The $\alpha$ parameter is highly correlated (R > 0.95) with the mean annual streamflow, suggesting that $\alpha$ equates to streamflow when *TWSA* is zero or close to zero for regions where the *TWSA* time series has no or limited trends. For example, Figure 11 shows that when *TWSA* is zero, the streamflow is approximately equal to the mean annual streamflow. While the exact interpretation of this point is difficult due to the monthly sampling of GRACE, the shaded boxes in Figure 11 highlight this concept. For example, between June and July 2010 and between November and December 2010, the *TWSA* transitions from positive and/or negative values. In both cases, the discharge also passes through the mean annual value (blue dashed line in Figure 11). While the slope of $\alpha$ and mean annual discharge forced through zero is 0.9, $\alpha$ is found to vary within ±25% (i.e., 0.75× to 1.25×) of the mean annual discharge at 68% of the gauges. Thus, $\alpha$ reflects the integration of a watershed's mean precipitation and the corresponding conversion of precipitation into streamflow during baseline conditions. This is evident in the relationship shown in Figure 5a and in the spatial pattern shown in Figure 4b. In regions, which receive more rainfall (i.e., corresponding to larger baseflow values), $\alpha$ is larger. Variations in this pattern are likely due to the differences in runoff production processes for individual watersheds. Thus, watersheds with similar precipitation patterns and runoff generation processes have similar $\alpha$ values.

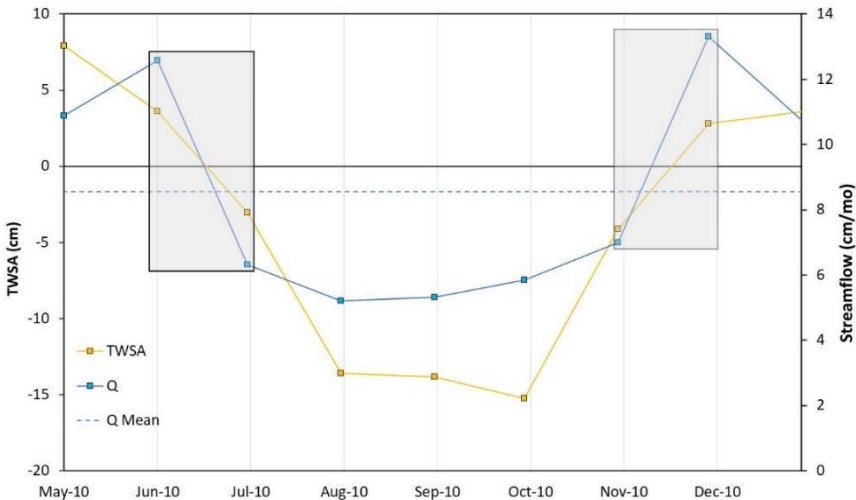

**Figure 11.** Time series of monthly *TWSA* and *Q* for one gauge (USGS ID 14316500), with the blue dashed line representing mean annual discharge; shaded boxes highlight the periods when *TWSA* passes through the zero-axis value, and discharge is nearly equal to mean annual discharge.

The exponential part of the equation ($\beta$) drives the streamflow higher or lower based on the TWS difference compared to the baseline period. This point is illustrated in Figure 12. For reference, the two model parameters are individually increased and decreased by 20%. When $\alpha$ is altered, the resulting *TWSA*–*Q* relationship is shifted up (increased $\alpha$) or down (decreased $\alpha$). However, when $\beta$ is altered, the resulting *TWSA*–*Q* relationship is rotated counterclockwise relative to the *TWSA* equal to 0 point (increased $\beta$) or clockwise (decreased $\beta$). The counterclockwise rotation (increased $\beta$) increases the discharges for positive *TWSA* values and decreases the discharges for negative *TWSA* values. The clockwise rotation (decreased $\beta$) decreases the discharges for positive *TWSA* values and increases the discharges for negative *TWSA* values. In other words, $\beta$ shifts the rate of runoff (sur-

face/subsurface) generation higher or lower for wet vs. dry conditions. Larger $\beta$ values imply more runoff production for wet conditions (larger positive *TWSA* values) as compared to dry conditions (more negative *TWSA* values). Smaller $\beta$ values imply more runoff production for dry conditions (more negative *TWSA* values) as compared to wet conditions (larger positive *TWSA* values). These differences likely account for watershed-specific processes related to groundwater (dry conditions) and surface/near-surface (wet conditions) runoff generation processes.

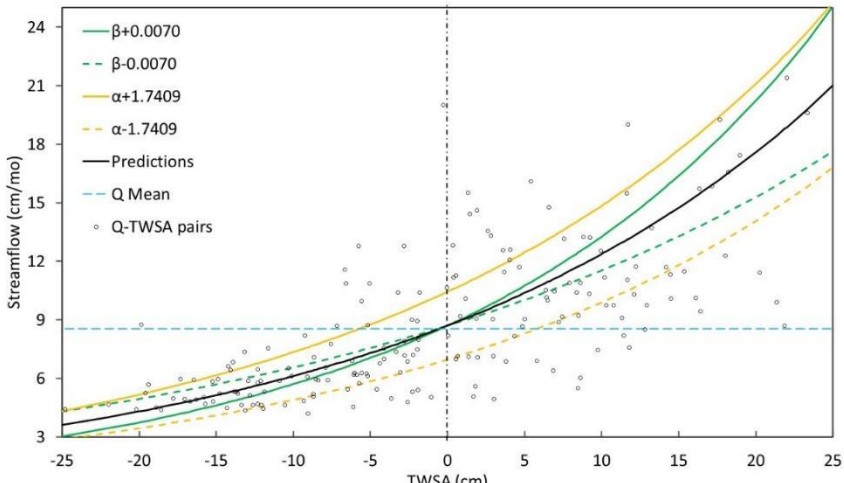

**Figure 12.** Observed streamflow vs. *TWSA* for one gauge (USGS ID 14316500) shown as black circles and the optimized model (black line); impacts of altering the optimized $\alpha$ (8.705) by $\pm20\%$ (yellow lines) and $\beta$ (0.0352) by $\pm$ 20% (green lines). Vertical dotted line represents *TWSA* 0; horizontal blue line represents mean annual discharge.

Building on the above model assessment, the mean streamflow and range in *TWSA* need to be relatively time invariant for these models to be applicable for preparing future discharge estimates (i.e., consistent with the training period). The $\beta$ variable is inversely correlated with the range in observed *TWSA* values (Figure 5b); it varies with the range in *TWSA*. While $\alpha$ tends to increase with mean monthly precipitation (and the associated streamflow) (Figure 5a), this is not always the case. For example, the streamflow does not always increase with precipitation in arid and semi-arid regions, e.g., when the soil water storage reservoirs are depleted in the summer, as observed in Ref. [48]. Hence, $\alpha$ does not depend on rainfall in these regions, and further research on regionalization is needed for semi-arid regions.

Geometric and topographic factors have also been observed to control the Q–S relation. Ref. [46] showed how the geometric factors of a catchment, such as the plan and profile shapes, affect the S–Q relations in simple and complex catchments using the kinematic wave theory. Here, only climatic and some hydrologic variables are linked with the location-specific model parameters ($\alpha$ and $\beta$). Thus, future research is needed to investigate how other catchment factors affect model parameters.

Snowmelt can be a significant source of streamflow in mountainous regions throughout the CONUS, e.g., in the Cascades, Sierras, Rocky and Appalachian Mountains. The influence of snowmelt on streamflow varies from rain on snow [49], local climatic trends and atmospheric circulation patterns [50]. Thus, eliminating a fixed number of months to optimize the model parameters is not perfect; a dynamic number of months each year could lead to the inclusion of more data pairs in the optimization process. Given that *TWSA* accounts for both inputs from precipitation and losses from snowmelt, our models inherently account for all runoff components and result in favorable model performance in many regions subjected to snow accumulation and melt.

Typically, only complex hydrologic models account for reservoirs and lakes in the modeling of streamflow. Some researchers have attempted to consider these effects using data-driven streamflow models [51]. Here, the proposed modeling approach performs poorly when the degree of water regulation is high. Future research is needed to quantify the degree of regulation needed to significantly impact model performance. This concept could also be used to identify months for which the proposed modeling approach can or cannot be used.

### 4.2. Model Training Periods

On average, the optimized models used 6 months of data and no lag. The sites with only 3–4 months of data in the training period account for roughly 30% of all sites. We observed that for nearly half (41%) of the gauges, December, January, February, March, April, May and June were excluded, with about a third (30%) of the gauges excluding December, January, February, March, April and May from their training periods. The percentage of gauges excluding a given month tends to increase from January to April and decrease thereafter, suggesting that snow accumulation and snowmelt do impact the exponential relation. For example, in a northeast USGS gauge 1053600 (Androscoggin River), where the *KGE* was 0.5 and the mean monthly snow cover was 35%, the optimized model was based on data for only five months—July, August, September, October and November—excluding the period, which was likely impacted by snow. Similarly, in a Pacific Northwest USGS gauge 12181000, where the *KGE* was 0.3 and the mean monthly snow cover was 47%, the optimized model was based on data for only three months—August, September and October—excluding the period, which was likely impacted by snow. Throughout the CONUS, the combination of months starting with December/January and ending with May/June are excluded the most, which is consistent with snow accumulation and melt periods in the CONUS.

The model period is also affected by reservoir regulation. For example, the optimized model for USGS gauge 03611500 (Ohio River) located downstream of the Kentucky Reservoir uses 3 months (August–October) of data in its training period. This is consistent with the Kentucky Reservoir releasing water at a controlled rate (i.e., not driven by changes in *TWSA*) from July to January based on a drawdown plan designed to make space for winter storms. Similarly, the Tennessee Valley Authority [52] stores streamflow from December to May and releases it as the downstream flows decrease throughout the remaining months. In Texas, the reservoirs used for flood control (e.g., Mansfield Dam) store streamflow during the monsoon/flood season (April–June), with little regulation during other months [53], which is reflected in the months excluded from the optimized models. In these examples and others, the proposed relations between *Q* and *TWSA* do not exist during the regulated months, which leads to those months being excluded from the optimization process.

### 4.3. Uncertainty

To determine the uncertainty estimates, only gauges with good *KGE* ($\geq 0.32$) and a reasonable number of *Q*–*TWSA* pairs available (at least 100) were considered. For these gauges, the data pairs (75% of the *Q*–*TWSA* data) from the optimized number of months were randomly sampled 100 times. For each sample set, an exponential model was fitted, and mean annual streamflow was estimated. The parameter uncertainty estimates were determined using the absolute percent difference between the optimized and median of the calibrated models' parameters (i.e., parameters from the 100 models based on randomly selecting 75 of the data pairs). For the mean annual streamflow, the uncertainty was determined using the absolute percent difference between the observed and median of the mean annual streamflow from the 100 calibrated models. The uncertainty in model parameters and mean annual streamflow estimated at each gauge location are reported in the published data for the optimized models [54].

In this study, the median and mean uncertainties for the optimized models at gauged locations are $\pm 5\%$ and $\pm 14\%$ for the alpha ($\alpha$) and $\pm 9\%$ and $\pm 15\%$ for the beta ($\beta$) parame-

ters, respectively. For the mean annual streamflow, the median and mean uncertainties are ±4% and ±9%, respectively, for the gauged locations throughout the CONUS. As noted above, for the variability in model parameters and the resulting discharge estimates, the uncertainty in the model parameters is reduced slightly when looking at simulated discharges (i.e., the discharge uncertainty is smaller) due to the two model parameter values changing in opposite directions.

### 4.4. Reasons for Poor Model Performance

For watersheds, where the *TWSA* series is modulated by GRACE signal leakage effects [21], snow accumulation/melt, water regulation, groundwater depletion, large lakes or trends in *TWSA* (e.g., depletion of lake levels), the exponential relationship does not capture the observed discharge patterns. For example, when the snow accumulates in a watershed, the *TWSA* increases, but there is no obvious increase in discharge (i.e., the snow accumulates without producing runoff). Similarly, when the snow melts, the discharge increases, but the *TWSA* tends to decrease. Neither case conforms with our exponential *TWSA–Q* relationship. Overall, 30% of the study gauges (99% of the gauges with *KGE* < 0.32) can likely be linked to one of the above-mentioned impacts. While the section below expands on these potential sources of poor model performance, it is challenging to separate these sources or identify the key sources leading to poor model performance. For example, many regions are likely impacted by multiple processes, such as snow accumulation/melt and water regulation (e.g., Missouri River Basin). Future research will focus specifically on identifying the dominant process impacting the *TWSA–Q* relationship.

Many watersheds in the southwest and western CONUS regions have negative trends in their *TWSA* time series (i.e., the slope of *TWSA* over time is <0). In these regions, the model performance is poor for 44% of the sites, accounting for roughly 54% of our study sites with *KGE* < 0.32. These gauge locations tend to have lower $\alpha$ values as well. The watersheds in these regions often have strict water regulations and dams, which can alter the hydrologic cycle and general relationship between the water storage and discharge. Figure 13a shows the streamflow for two gauges: upstream and downstream of a reservoir. Clear impacts of regulation (i.e., increases in downstream discharge with no associated increases in upstream discharge, followed by a decrease in reservoir storage) are visible in April 2018, June 2019 and August 2020.

As discussed previously, *TWSA* clearly varies with snow cover in regions where runoff depends mostly or partly on snowmelt. Overall, half of the watersheds with *KGE* < 0.32 are impacted by snow; these sites have median monthly snow cover greater than 30%. This phenomenon was observed in upper Mississippi, Colorado and New England. An example from New England is highlighted in Figure 13b, where *TWSA* increases with increasing snow cover, while the streamflow remains relatively constant.

In Michigan, the *TWSA* varies with lake Michigan–Huron water level, which is likely due to the leakage effects, as has been noted by other researchers. The GRACE leakage effects are caused by spectral truncation, as well as the particular GRACE orbit geometry resonance during the determination of sectorial GRACE Stokes coefficients and the filtering of GRACE observations [21]. Model performance is generally poor in these locations because watershed-averaged *TWSA* is impacted by nearby lake water mass changes more than as a result of watershed storage changes. Almost 76% of the sites in Michigan have a *KGE* of less than 0.32, accounting for 6% of our study sites with a *KGE* of less than 0.32. The trend in the *TWSA* time series is 0.25 to 0.35 (cm/month) for catchments in Michigan, which is the highest observed trend in the CONUS. In Figure 13c, the *TWSA*, lake water level and streamflow of one such gauge are highlighted to show how *TWSA* clearly varies with lake level, and the discharge does not follow a similar pattern.

Roughly 43% of the watersheds in Florida have a *KGE* of less than 0.32, which represents 3% of our total number of sites with a *KGE* of less than 0.32. This may be linked to *TWSA* in the southeast CONUS being impacted by El Niño conditions—a phenomenon observed in Ref. [55], which showed larger *TWSA* magnitudes in El Niño vs. La Niña

years due to precipitation variability. However, additional research is needed to better understand the potential reasons (e.g., proximity to the Gulf of Mexico and the Atlantic Ocean) behind poor model performance in Florida.

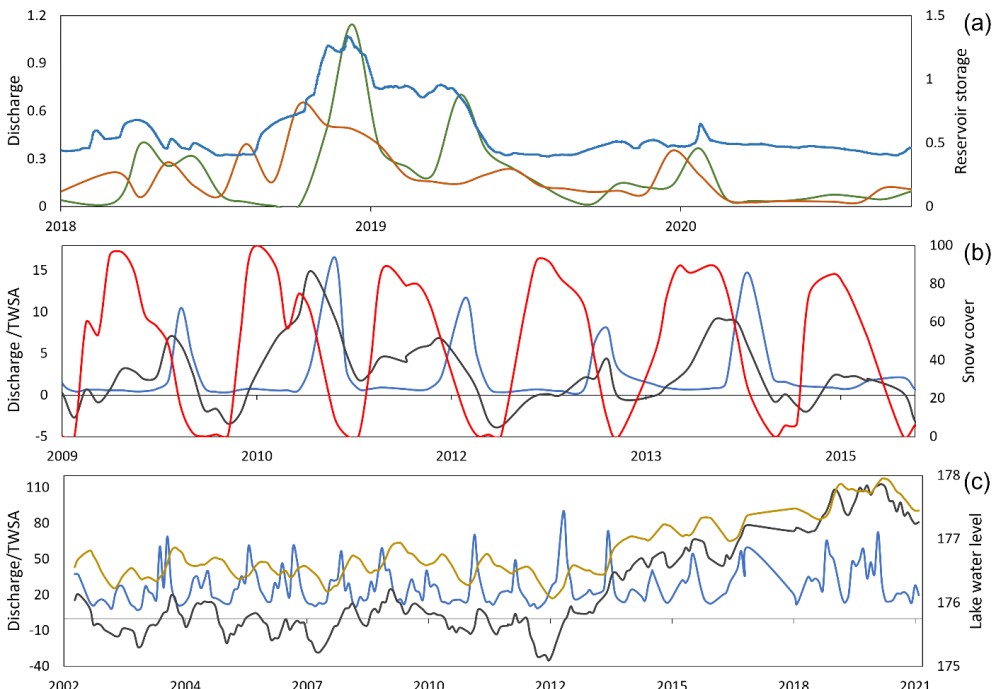

**Figure 13.** (**a**) Streamflow for USGS gauges upstream (6856600) (orange) and downstream (6857100) (green) of a reservoir with *KGE* values of 0.46 and 0.16, respectively, left *y*-axis (cm/month), and Milford reservoir water level (cubic km), right *y*-axis (blue); (**b**) streamflow (cm/month) (blue) and *TWSA* (black) on left *y*-axis and snow cover (average monthly % snow cover) (red) on right *y*-axis for USGS gauge 6207500 with a *KGE* of 0.21; and (**c**) Michigan–Huron lake water level (m) (yellow) on right *y*-axis and *TWSA* (cm) (black) and streamflow (10× cm/month) (blue) on left *y*-axis for USGS gauge 4154000 with a *KGE* of 0.03.

Our understanding of when the proposed modeling approach does not work well and the selection of months used for parameter optimization can be used to assess locations for which the model predictions are applicable for decision making. For example, in watersheds influenced by upstream dams, the proposed relationship does not apply for months, which are known to be highly regulated. The optimized models are only trained during unregulated seasons; using predictions for regulated months is not applicable. For watersheds, where the *TWSA* is influenced by El Niño conditions or nearby lake or sea levels, or where the *Q–TWSA* relationships are impacted by snow accumulation/melt, additional steps are needed to adjust *TWSA* signals or de-seasonalize the data prior to regionalization.

### 4.5. Comparison with Previous Models

The models presented in this study generally perform more favorably as compared to the previously developed model presented in Ref. [6] (Figure 14a). The *KGE* for the previous models were determined from the published model parameters, which systematically excluded 5 months (November–April). The data of the previously published model was extracted from Figure 3 in Ref. [6]. Here, an optimization approach was used to determine which months of data should be used, which explains the improvement in model performance. The results show (Figure 14b) that the models developed here excluded a mix of months, which likely included snow accumulation/melt and reservoir regulation. For example, at USGS gauge ID 7374000, the optimized model excluded December–July,

which included periods impacted by snow and upstream reservoir releases during the fall (as discussed in Section 3.3). The optimization process selects the model, which has identified the months in which the proposed relation exists. The decrease in *KGE* for USGS gauge ID 7374000 is minimal (4.2%) and may be a result of including a longer data record in the optimization process used here. The key finding from this comparison is that the optimization process generally improves the model performance over forcing a fixed data period, which is similar to the comparison between the baseline and optimized models in this study.

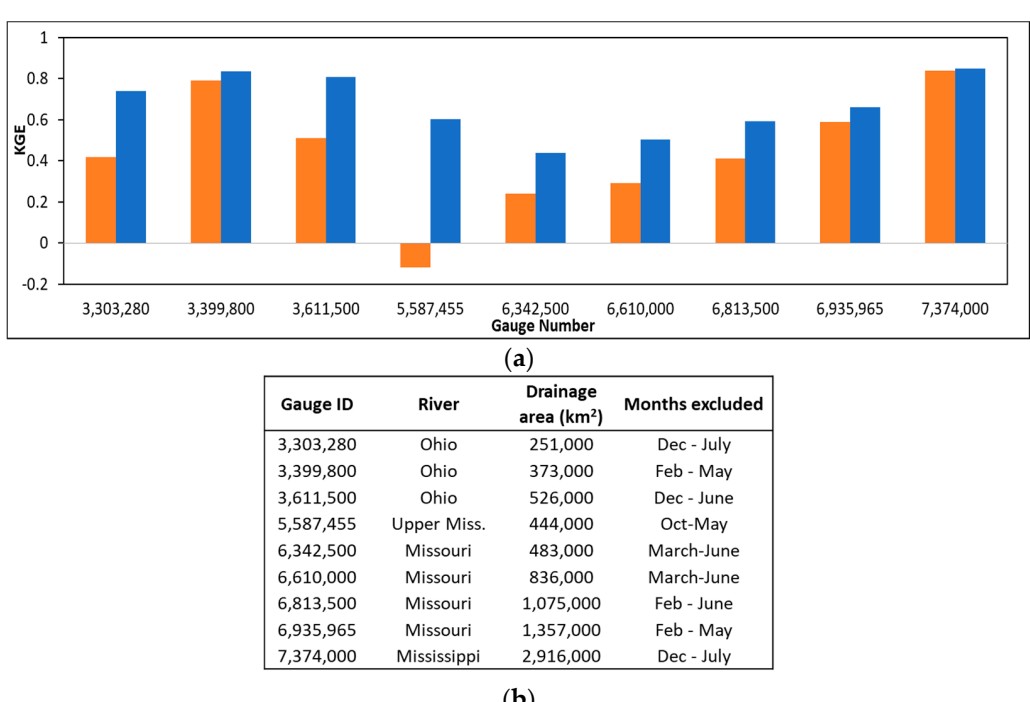

(a)

| Gauge ID | River | Drainage area (km²) | Months excluded |
|---|---|---|---|
| 3,303,280 | Ohio | 251,000 | Dec - July |
| 3,399,800 | Ohio | 373,000 | Feb - May |
| 3,611,500 | Ohio | 526,000 | Dec - June |
| 5,587,455 | Upper Miss. | 444,000 | Oct-May |
| 6,342,500 | Missouri | 483,000 | March-June |
| 6,610,000 | Missouri | 836,000 | March-June |
| 6,813,500 | Missouri | 1,075,000 | Feb - June |
| 6,935,965 | Missouri | 1,357,000 | Feb - May |
| 7,374,000 | Mississippi | 2,916,000 | Dec - July |

(b)

**Figure 14.** (**a**) Comparison between optimized (i.e., best) model from this study (blue bars) and previously published results using similar concepts (orange bars) [6] and (**b**) key information on the USGS gauges used in the analysis; months excluded, such as December–March, means that the *Q–TWSA* data pairs for December, January, February, March were excluded from the model training period.

Extending our analysis to larger rivers, we looked at the export of water from select rivers draining to the ocean. For these rivers, the estimated discharges using the preliminary regionalized models are compared with gauge estimates (Table 1). In general, the percentage difference is less than 25% for all rivers with an average absolute difference of 14%. The largest differences are over-estimations for San Joaquin (24%) and Susquehanna (23%) rivers. For the period 2002–2021, the net export in total discharge was over-estimated by 6%.

In another related study (Ref. [56]), gridded runoff and baseflow were estimated with the water balance method using satellite data. The NCEP-NARR precipitation and PY-CRAE ET was resampled to GRACE *TWSA* resolution (0.5 degrees), and the baseflow index map was used for baseflow estimation. They evaluated model performance in 18 USGS HUC basins distributed throughout the CONUS and reported a RMSE and $R^2$ of 14.8 mm/month and 0.81, respectively, when estimating the annual mean. This demonstrated that the use of satellite data in physical models shows promising results. The mean annual runoff predicted from the optimized models in this study, while not examining exactly the same HUC basins, has a RMSE and $R^2$ of 6.2 mm/month and 0.99, respectively.

**Table 1.** Estimated streamflow (SF) to the ocean for select rivers in CONUS; actual and predicted discharges for major rivers.

| River | Drainage Area | Observed SF (cm/mon) | Predicted SF (cm/mon) |
|---|---|---|---|
| Mississippi | 2,960,065.51 | 1.49 | 1.35 |
| Columbia | 652,024.38 | 2.58 | 2.47 |
| Colorado | 619,277.82 | 0.02 | 0.02 |
| Riogrande | 491,295.94 | 0.01 | 0.01 |
| Brazos | 113,940.98 | 0.54 | 0.65 |
| Sacramento | 71,243.45 | 2.36 | 2.63 |
| Susquehanna | 71,048.51 | 4.60 | 5.65 |
| San Jaoqin | 44,805.09 | 0.72 | 0.89 |

## 5. Conclusions

Streamflow prediction from GRACE terrestrial water storage anomalies (*TWSA*) is a relatively new concept. Here, non-linear models were developed for predicting monthly streamflow from monthly *TWSA* series at 2870 USGS streamflow gauge locations. The model performance at 70% of the study sites exceeds the levels expected for other remotely sensed discharge approaches (*KGE* $\geq$ 0.32).

The *Q–TWSA* relationships were optimized by exploring the applicable *Q–TWSA* data pairs (i.e., excluding select months of data) and lag between data pairs. We showed that the parameters for the top-performing models vary somewhat but that the net effect in predicted mean discharges is muted; optimized model parameters ($\alpha$, $\beta$) are within 30% of the baseline model parameters at 87% of the sites for $\alpha$ and 59% of the sites for $\beta$, while the predicted mean discharges are within 30% of the baseline model predictions at 99% of the sites. Thus, the selection approach used did not overly influence the optimized model parameters or performance. Additionally, the results show that the optimization process improved model performance as compared to a previous study.

The model parameters ($\alpha$, $\beta$) are shown to be related to annual precipitation and the range in *TWSA* with $R^2$ of 0.4 and 0.5, respectively. Considering the spatial extent of GRACE *TWSA* measurements and the methodology involving drainage network analysis applied in this context, it is observed that the *TWSA* signals exhibit resemblances along major rivers. This similarity results in comparable $\beta$ values, while the $\alpha$ values tend to increase along the river network to accommodate the escalating discharge corresponding to the expanding drainage area along these rivers.

The model performance (*KGE* > 0.32 or *KGE* < 0.32) can be predicted based on the correlation between precipitation and *TWSA*, the trend in *TWSA* over time, the magnitude of annual P/PET and the correlation between snow cover and *TWSA*, where all values are determined from remotely sensed data products. A simple decision tree for classification utilizing the average annual precipitation relative to potential evapotranspiration (P/PET), along with the correlation of *TWSA* with precipitation and snow cover, proved sufficient for predicting the applicability of the model form, with a hit score of 93%.

Across the continental United States (CONUS), the time frame spanning from December/January to May/June is the most frequently excluded (i.e., the months where *TWSA–Q* relationships are not found). This pattern generally aligns with the periods of snow accumulation and melting within the CONUS. While sites with poor model performance are often associated with snow-dominated systems, other factors, such as trends in *TWSA* and water regulation, are also found to contribute to a lack of model performance. Future efforts will investigate the methods for isolating the dominant cause of poor model performance.

The overall results suggest that expansion to other regions and ungauged locations has potential. While not intended for event-type modeling, the proposed modeling approach

has the potential to provide new insights on the annual, seasonal and baseflow discharge characteristics. Given the reliance on only two parameters and GRACE *TWSA*, the insights gained from the proposed approach offer information, which can be further exploited (e.g., calibration/validation) in hydrologic models, which are able to simulate physically based rainfall–runoff processes or altimetry-based remote-sensing discharge approaches.

**Supplementary Materials:** The following supporting information can be downloaded at: https://www.mdpi.com/article/10.3390/rs15184516/s1, Figure S1: A watershed in lower Mississippi inside one single pixel and 3 gauges with different Drainage area. The hydrograph and Q vs TWSA graph are plotted; Figure S2: Maps of the following A. The KGE of the optimized model on all the 12 months. B. KGE of the optimised models on the subsetted (optimized) months. The subsetted months are the months during which the exponential relation exists between TWSA and discharge. C. The difference between these two types of KGE.

**Author Contributions:** Conceptualization, E.B.; methodology, E.B. and B.D.; software, E.B.; validation, B.D. and E.B.; formal analysis, B.D.; investigation, B.D.; resources, E.B.; data curation, B.D.; writing—original draft preparation, B.D.; writing—review and editing, E.B. and B.D.; visualization, B.D.; supervision, E.B.; project administration, E.B.; funding acquisition, E.B. All authors have read and agreed to the published version of the manuscript.

**Funding:** This study was supported by NASA GRACE-FO Science Team award #80NSSC20K0742.

**Data Availability Statement:** The optimized models along with the results of uncertainty analysis and processed GRACE TWSA times series used in the study are available at a CUAHSI repository [54], accessible to readers via the link.

**Conflicts of Interest:** The authors declare no conflict of interest.

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
