# Peer review of "Estimating Monthly River Discharges from GRACE/GRACE-FO Terrestrial Water Storage Anomalies"

_remotesensing, doi:10.3390/rs15184516_

Round 1

Reviewer 1 Report

The manuscript topic is interesting and up-to-date. I appreciate this research which combines many data sources into one model. I only have a few suggestions and comments:

1. References in the text are not unified. Usually there is a number in square brackets but sometimes there is a name and year without a number which is confusing. I would suggest to unify the method. Moreover I did not find some references in the text [34-37] and [45-46].

2. I miss any reference to Kling Gupta Efficiency (KGE).

3. From description of the experiment (using training and testing data) it evokes that some machine learning method has been applied. If yes, I would recommend to state it explicitly in the text or even use the keyword "machine learning".

4. The sentence on lines 53-55 "The GRACE observations are ultimately used to derive monthly TWSA, ... relative to an average potential for the period 2004-2009." is misleading. It depends on which reference model is subtracted. The reference does need to be the period 2004-2009. It depends on specific product or it depends on user, which reference model user subtracts. So I would suggest to finish the sentence "... relative to chosen reference period, in our case the period 2004-2009."

5.  The sentence on lines 97-98 is probably misleading. Missing GRACE values in 2017-2018 period is likely due to the gap between GRACE and GRACE-FO missions and not due to battery related issues.

1. Some parts of the text repeat almost word-by-word the same information. Such duplicities should be avoided (e.g. lines 144-147 repeats information from lines 75-78).

2. Line 440 (page 12), it should be "... streamflow is approximately equal ..." not equals.

3. Line 574 (page 16), km2 - 2 should be the upper index.

Author Response

Comment 1: References in the text are not unified. Usually there is a number in square brackets but sometimes there is a name and year without a number which is confusing. I would suggest to unify the method. Moreover I did not find some references in the text [34-37] and [45-46].

Response: The use of name and year citations at beginning of sentences was replaced with [#], and all references are now cited in the text.

Comment 2: I miss any reference to Kling Gupta Efficiency (KGE).

Response: KGE reference is added: Gupta, H. V.; Kling, H.; Yilmaz, K.K.; Martinez, G.F. Decomposition of the Mean Squared Error and NSE Performance Criteria: Implications for Improving Hydrological Modelling. J Hydrol (Amst) 2009, 377, 80–91, doi:10.1016/j.jhydrol.2009.08.003.

Comment 3: From the description of the experiment (using training and testing data), it evokes that some machine learning method has been applied. If yes, I would recommend to state it explicitly in the text or even use the keyword "machine learning".

Response: No machine learning technique was used. In this study, the term “training data” corresponds to the data used to fit the models (i.e., original data minus any months of data excluded). Text was added to clarify.

Comment 4: The sentence on lines 53-55 "The GRACE observations are ultimately used to derive monthly TWSA, ... relative to an average potential for the period 2004-2009." is misleading. It depends on which reference model is subtracted. The reference does need to be the period 2004-2009. It depends on specific product or it depends on user, which reference model user subtracts. So I would suggest to finish the sentence "... relative to chosen reference period, in our case the period 2004-2009."

Response: We used the JPL mascon products for Terrestrial Water Storage Anomaly (TWSA) data. The data product processing documentation states that the time-mean baseline for the final values is 2004-2009. https://grace.jpl.nasa.gov/data/get-data/jpl_global_mascons/. Text was modified to clarify this point.

Comment 5: The sentence on lines 97-98 is probably misleading. Missing GRACE values in 2017-2018 period is likely due to the gap between GRACE and GRACE-FO missions and not due to battery related issues.

Response: The text was modified to reference the missing data between missions.

Comment 6: Some parts of the text repeat almost word-by-word the same information. Such duplicities should be avoided (e.g. lines 144-147 repeats information from lines 75-78).

Response: The text in the Introduction was simplified and the text in the Methodology was expanded to eliminate the duplication. The paper was also reorganized to eliminate the need to duplicate content.

Reviewer 2 Report

Please find the attached pdf for detailed comments and suggestions. 

Reviewer 3 Report

Review of manuscript  for Remote Sensing (MDPI)"Estimating monthly river discharges from GRACE/GRACE-FO total water storage anomalies"by Bhavya Duvvuri & R. Edward Beighley

August, 2023

GENERAL
The authors propose an approach to predict the monthly streamflow from GRACE Total Water Storage anomalies and water discharge - weighted by the drainage (sub)surface - at nearly 2 970 river gauge stations in the US forming a dense network of measurements.  The empirical exponential model is simply based on two parameters to be fitted from the combination of GRACE data and the river discharge variations recorded at each station. The determination of these two local parameters that are used to characterize the river flow suffers from surface data resolution and spacing, the considered time interval of computation, and often the delayed water storage caused by (human) dam or snow. Reconstruction of river discharge maps are also shown and provide realistic estimates of river discharges over the US. My major comment: Besides the demonstration is clear and detailed, the Conclusion is relatvely short and it needs to be much completed by results (and comments) to highlight relevant sensitity tests and issues of the method. Presentation of the results (before Discussion) could have been much structured and simplied for clear understanding. Therefore I consider the authors should do last efforts to make their very interesting work worth for publication.

SOME MINORS
P.1, L.10: Simulating River discharge ---> Simulating river discharge
P.1, L.15: the Continental U.S. ---> the continental U.S.
P.5, L.214: Please indicate what these gauge stations are.
P.11, L.406: water year This ---> water year. This
P.14, L.500: the propose modeling ---> the proposed modeling
P.14, L.507: Potential Evapotranspiration (P/PET) ---> Potential EvapoTranspiration (P/PET)
P.14, L.514: Potential Evapotranspiration ---> Potential EvapoTranspiration
P.17, L.609: timeseries ---> time series
P.17, L.625: GRACE Leakage effect is ---> GRACE leakage effects are
P.17, L.625: Leakage effects are not only due to pure spectral trunctation but also the particular GRACE orbit geometry resonance during the determination of sectorial GRACE Stokes coefficients.
P.18, L.637-638: El Nino ---> El Niño       La Nina ---> La Niña
P.20, Conclusion: there are repetitions about the novelty of the proposed empirical approach.
P.20, Conclusion, L.715-716: repetition of "require/required". Please rephrase.
In all the document, "in-situ" should be "in situ".
In all the text, the authors citation should be numbered instead of, e.g. "Kirchner-2009" to be cosnsistent with the citation format.

--- End of document ---

The manuscript requires a moderate English check up.

Round 2

Reviewer 2 Report

I am currently traveling and cannot access the review form properly. However, I quickly read through the changes and authors responses. Although there can be further iterations for improvement, I suggest accepting the manuscript in its current form.

Author Response

Thank you for your insights.